# Sleep Quality among the Elderly in 21st Century Shandong Province, China: A Ten-Year Comparative Study

**DOI:** 10.3390/ijerph192114296

**Published:** 2022-11-01

**Authors:** Zenghe Yue, Yi Zhang, Xiaojing Cheng, Jingxuan Zhang

**Affiliations:** 1Shandong Mental Health Center, Shandong University, Jinan 250014, China; 2Shanghai Key Laboratory of Psychotic Disorders, Shanghai Mental Health Center, Shanghai Jiaotong University School of Medicine, Shanghai 200030, China

**Keywords:** sleep quality, aging, logistic model, China

## Abstract

Background: Despite the enormous changes observed in China since entering the 21st century, little is known about changes in sleep quality among older adults. Aims: The purpose of this study is to explore the changes, features, and influence factors of sleep quality among the elderly in a ten-year period, providing evidence for sleep-quality enhancement. Methods: The data were obtained from the data of epidemiological sampling surveys on mental disorders in Shandong province in 2004 and 2015. A total of 4451 subjects (aged ≥ 60 years) in 2004 and 10,894 subjects (aged ≥ 60 years) in 2015 were selected by the multistage stratified sampling method. The demographic information and Pittsburgh Sleep Quality Index (PSQI) were collected. Results: The adjusted 1-month prevalence of poor sleep in 2015 was 22.5% (95% CI:21.7–23.3), which is lower than that in 2004 (24.8%) (95% CI:23.5–26.0, *p* = 0.002). The total score of the PSQI in 2015 (4.74 ± 3.96) was lower than that in 2004 (4.97 ± 4.18, *p* = 0.002). In 2015, a binary multi-factor logistic and stepwise regression analysis showed that being female, living in a rural area, living alone, being older, spending less years in school, and being jobless/unemployed made the participants more likely to develop poor sleep (*p* < 0.05, *p* < 0.01). Conclusions: In 2015, the overall sleep quality of the elderly (aged ≥ 60) in Shandong province was significantly improved compared to 2004. After more than 10 years, the characteristics of the elderly with sleep disturbances in Shandong province has changed. Therefore, more attention should be paid to gender, location of residence (rural or urban), living arrangement, age, education, occupation, and other factors to improve the sleep quality of the elderly.

## 1. Introduction

Since the beginning of the 21st century, global aging has become an increasing problem, and older adults’ physical and mental health has become a priority in public health today [1,2]. By 2050, about 80% of older people aged 60 and above will live in low- and middle-income countries, and the elderly population will grow faster than in high-income countries [3]. The problem is particularly acute in China, the most populous country. In today’s China, the proportion of the elderly (aged 60 and above) has risen from 10.33% in 2000 to 18.70% in 2020 [4]. The population of the elderly in China reached 264 million in 2020. In Shandong province, the second-most populous province in China, the proportion of the population of elderly comes to 20.90% [4]. This trend in aging is increasingly aggravated.

The physical and mental health of the elderly plays a vital role in public health and social burden, in which sleep problems have a particularly prominent impact on the elderly. The overall prevalence of poor sleep among the elderly in China has reached 35.9% [5]. Several studies have shown that a long-term lack of sleep in the elderly can lead to decreased immunity and various physical and mental disorders, such as neurasthenia and gastroesophageal reflux disease [6]. It can even increase the risk of high blood pressure, heart disease, depression, Alzheimer’s disease, and suicidal behavior [7,8,9,10]. Good sleep is essential for individuals to regulate their mood and for their perception of subjective well-being [11,12]. In addition, sleep is significantly associated with social, physical, and mental health, as well as overall quality of life [11,13]. Another study suggests that insomnia plays an important role in mediating the association between social media use and subjective well-being in today’s widespread use of social media [14].

Since China has entered the 21st century, there has been a rapid development in all aspects, with sweeping economic and lifestyle changes, which have had a major impact on the elderly population. Little is known about the changes in the sleep quality of the elderly population. This study consists of two parts, with two surveys spanning ten years, to maximize the response to changes in sleep quality in the elderly population since 2000. It provides suggestions for the precise targeting of sleep quality in the elderly population. It is one of the critical projects in the epidemiological sample survey of mental disorders in Shandong province in 2004 and 2015. The Pittsburgh Sleep Quality Index scale was used as the primary research tool to conduct a longitudinal and cross-sectional comparative study of the data from two epidemiological surveys of mental disorders in Shandong province in 2004 and 2015. This study aims to understand the current situation of sleep quality among people aged ≥ 60 years in Shandong province and investigate the changes, characteristics, and related influencing factors of sleep quality, providing a reference for improving the sleep quality of the elderly in Shandong province.

## 2. Methods

### 2.1. Study Population

The present study consists of two parts, and the information was obtained from data from the Shandong Province Mental Disorders Epidemiological Survey (SMDES) in 2004 and 2015 [15]. Since entering the 21st century, two province-wide epidemiological surveys on mental disorders have been conducted in Shandong province (2004 and 2015). Both surveys used the same sampling method, research instruments, and diagnostic criteria, with a reasonable study design and reliable findings [15]. In addition, China underwent dramatic changes within this decade, such as dramatic socioeconomic improvements. Using data from two time points can be a good way to explore the impact of economic development on the health of the elderly. In 2004, a multistage stratified whole-group sampling method was used to randomly select 10 streets and 30 townships in 5 districts and 15 counties of 5 prefecture-level cities in Shandong province, with a total of 20 neighborhood committees and 60 administrative villages. There were 4482 older adults aged ≥ 60 years, including 2 with dementia, 3 with severe deafness, and 26 with incomplete data information. Finally, 4451 older people were included in this study, and the effective rate was 99.3%. In 2015, a multistage stratified whole-group sampling method was used to randomly select 34 streets and 62 townships in 49 counties (cities and districts) of 16 cities and a total of 92 villages (neighborhood) committees in Shandong province. There were 11,155 older adults aged ≥ 60 years, including 27 with dementia, 33 with severe deafness, and 201 with incomplete data information. Finally, 10,894 elderly people were included in this study, with an effective rate of 97.7%.

Overall, 15,345 older adults were included in this study, with a mean age of 68.43 ± 6.62 years for the 2004 population and 68.90 ± 6.99 years for the 2015 population. Written informed consent to participate was obtained, and the Shandong Mental Health Institutional Review Board approved the study (2022 (R) Ethics Review No. 39). The 2004 SMDES was conducted from November 2004 to March 2005, and the 2015 SMDES was conducted from September 2015 to December 2015. Due to population mobility and aging, the differences in population composition between the 2015 and 2004 populations were statistically significant between all groups, except for the gender factor. It is worth noting that the health insurance category changed dramatically due to changes in China’s healthcare policy. The proportion of the self-pay population decreased from 67.69% in 2004 to 3.09% in 2015, while the proportion of the elderly with new agricultural cooperative insurance increased from 14.29% to 74.32% (Table 1).

### 2.2. Study Tools

Demographic information was collected, including gender, age, years of education, marital status, mode of residence, occupation, medical source, and physical illness. Previous studies found that the influencing factors associated with poor sleep include the above basic characteristics, such as gender and age [5]. Our collection of detailed basic information on the population helps to build a more valid and complete prediction model.

The Pittsburgh Sleep Quality Index (PSQI) scale [16] was developed by Dr. Buysse and collaborators, and the Chinese version was translated and tested by Liu and collaborators [17]. It has good reliability and validity and is one of the most widely used standardized indicators for assessing sleep quality. The scale applies not only to patients with sleep disorders, but also to the general population in evaluating sleep quality. The PSQI is used to assess the sleep quality of subjects in the previous one month and consists of seven factors: subjective sleep quality; sleep latency; sleep duration; sleep efficiency; sleep disturbance; sleep medication use; and daytime dysfunction. Each factor is scored on a scale of 0 to 3, from low to high frequency, and the total PSQI scores range from 0 to 21. In the Chinese population, a total PSQI score of 7, rather than 5, as the threshold tends to have a higher sensitivity and specificity in identifying poor sleep quality [17,18,19], so the present study used a total PSQI score > 7 to consider the presence of poor sleep quality.

### 2.3. Study Design

To understand the current status of sleep quality among older adults in Shandong province and the changes in sleep quality among older adults in Shandong province after entering the 21st century, we compared two province-wide epidemiological surveys of sleep quality among older adults in Shandong province (2004 and 2015) [15]. This included a comparison of the prevalence of poor sleep, a comparison of sleep quality (PSQI), a comparison of sleep-related influencing factors, and the development of a predictive model for poor sleep. In both surveys, we used the same questionnaire to collect detailed general information about the population, including gender, age, area of residence, educational level, and marital status. Additionally, the same research tools and diagnostic criteria were used to diagnose poor sleep. When comparing the prevalence of poor sleep, we standardized the prevalence of poor sleep using the data from the seventh national census of China (2020) as the standard population to eliminate the differences in age structure in the two surveyed populations. In addition, a validated prediction model was necessary to intervene accurately in the sleep of older adults. Therefore, we established binary logistic regression equations with the presence or absence of poor sleep (0 = none, 1 = yes) as the dependent variable and general information as the independent variable. Based on the logistic regression equation from 2015, we built the nomogram prediction model. This model not only identifies whether there was any change in the influencing factors related to sleep quality between the two surveys, but also provides a reference for precise intervention in the sleep of the elderly. More importantly, the results of our study can provide a reference for the next survey.

### 2.4. Statistical Analyses

The 1-month prevalence of poor sleep quality by sex, age, and area of residence was calculated among the elderly in Shandong province, China, and the prevalence was compared by applying chi-square tests. Because of the inconsistent age structure between the 2004 and 2015 samples, we used data from the seventh National Census of China in 2020 as the standard population to adjust the prevalence. We applied t-tests and one-way ANOVA to independent samples to discuss the differences between the two surveys in actual mean sleep duration by gender, age, and region of residence. Then, we compared the differences between the PSQI scores for each entry and the total score between 2004 and 2015. We conducted a multi-factor logistic and stepwise regression analysis for both 2004 and 2015, and we plotted the nomogram prediction model and the calibration curves for the 2015 information based on the logistic regression results. We then put the 2004 data into the 2015 nomogram prediction model and plotted its calibration curve to observe the changes.

A database was created using Epidata and Excel for double data entries. The statistical analysis was performed by SPSS 26.0 and R 3.4.2 software. A two-tailed significance, defined as *p*-value < 0.05, was used for all analyses. The Benjamini and Hochberg FDR method was applied for multiple comparisons.

## 3. Results

### 3.1. Prevalence of Poor Sleep Quality

A total of 1070 (24.0%, 95% CI:22.8–25.3) out of 4451 people had poor sleep quality in 2004, and 2390 (21.9%, 95% CI:21.2–22.7) out of 10,894 people had poor sleep quality in 2015. The overall population-adjusted 1-month prevalence of poor sleep in 2015 was 22.5% (95% CI:21.7–23.3), lower than that of the 24.8% (95% CI:23.5–26.0, *χ*^2^ = 9.33, *p* = 0.002) prevalence in 2004 using age stratification of the population aged 60 years or older in the *China Statistical Yearbook 2021* [20] (information from the seventh national census in 2020). Due to the presence of fractions and other circumstances in the calculation, there may be a small error in summing the number of cases (Table 2). The adjusted prevalence characteristics of poor sleep were approximately the same between 2004 and 2015 by gender and age, with a higher prevalence among women than men (*p* < 0.001) and a higher prevalence related to higher age (*p* < 0.001). It is noteworthy that the difference between regional urban and rural areas was not statistically significant in 2004, but the difference between regional urban and rural areas was statistically significant in 2015, with urban areas being lower than rural areas (*χ*^2^ = 15.46, *p* < 0.001), as shown in Table 3.

### 3.2. Average Sleep Duration

The actual mean sleep duration in 2015 was 7.39 ± 1.88 h, longer than the actual mean sleep duration in 2004, which was 7.22 ± 1.77 h (*t* = −5.76, *p* < 0.001). The characteristics of the actual mean sleep duration in 2004 and 2015 were approximately the same, both being longer in males than in females, in rural areas than in urban areas, and in lower age groups than in higher age groups, all with statistically significant differences (*p* < 0.001). The actual mean sleep duration was longer in 2015 than in 2004 in the same population (*p* < 0.05, *p* < 0.01), and the results are shown in Table 4.

### 3.3. PSQI Scores

The total PSQI score in 2015 was lower than in 2004 (2015: 4.97 ± 4.18; 2004: 4.74 ± 3.96; *t* = 3.19, *p* = 0.002). There were no significant differences between 2015 and 2004 in sleep medication use and subjective sleep quality (2015: sleep medication use: 0.11 ± 0.53, subjective sleep quality: 1.00 ± 0.70; 2004: sleep medication use: 0.11 ± 0.51, subjective sleep quality: 0.97 ± 0.74). Among the remaining factors, the 2015 scores for sleep latency 1.02 ± 1.01, sleep duration 0.54 ± 0.95, and sleep efficiency 0.90 ± 1.17 were lower than the 2004 scores for sleep latency 1.07 ± 1.01, sleep duration 0.86 ± 1.01, and sleep efficiency 1.09 ± 1.22 scores. The 2015 scores for sleep disturbance 0.69 ± 0.63 and daytime dysfunction 0.48 ± 0.84 were higher than the 2004 scores for sleep disturbance 0.49 ± 0.56 and daytime dysfunction 0.37 ± 0.76. The differences in the above results were all statistically significant, as shown in Figure 1.

### 3.4. Multi-Factorial Analysis of Sleep Quality

The dichotomous multi-factor logistic and stepwise regression analysis was performed with the presence or absence of poor sleep quality as the dependent variable (0 = none, 1 = yes) and the different variables in the general data as the independent variables for 2004 and 2015, respectively, as shown in Figure 2 and Figure 3. The results showed that, after 10 years of changes, the regression model of sleep quality for people aged 60 years and above in Shandong province changed. Gender, age, occupation, source of medical care, and marriage were the influencing factors of sleep quality in 2004. Females and people of a higher age, as well as people who were jobless and unemployed, had new cooperative medical care, and were unmarried/divorced were more likely to suffer from poor sleep (*p* < 0.05, *p* < 0.01). In 2015, gender, living area, living arrangement, age, education, and occupation were the predictive factors of sleep quality. Females and people who lived in a rural area, or lived alone, or those of a higher age, with low years of study, and those who were jobless and unemployed people were more likely to suffer from poor sleep (*p* < 0.05, *p* < 0.01). Based on the logistic regression results, the nomogram prediction model and its calibration curve were plotted for the 2015 information, where C-index = 0.636. We then put the 2004 data into the 2015 prediction model and plotted the calibration curve. The results show that the C-index = 0.617, but the calibration test failed (U = 0.004, S: *p* = 0.000). The 2015 prediction model is no longer suitable for 2004, as shown in Figure 4.

## 4. Discussion

The increasing aging of China’s Shandong province since the 21st century has made the elderly a huge group that cannot be ignored, and sleep problems are prevalent in the elderly population [21]. In this study, we compared the sleep quality of the elderly population in Shandong province in 2004 and 2015. It was found that the prevalence of poor sleep quality among older adults in Shandong province in 2015 was lower than that in 2004, and the actual sleep duration in 2015 was longer than that in 2004. Moreover, after more than 10 years, different factors contributed to sleep quality in Shandong province in 2015 compared to 2004. Additionally, we can see that, when putting the 2004 data into the 2015 prediction model for validation, it does not pass the calibration degree test, indicating that the factors influencing the sleep quality of older people in Shandong province have changed.

Overall, sleep quality was better in 2015 than in 2004. However, in 2015, older adults suffered more sleep disturbances and daytime functioning. One reason for this may be the spike in housing prices over the past decade or so and because most children cannot solve housing and other problems on their own, turning to the older adult population [22]. This has increased the pressure on the older age group to consider more aspects or to continue working intensely during the day to face these problems [22]. This aggravates the vicious cycle of overworking in older people, leading to poor sleep quality at night and fatigue during the day, so some PSQI factors scores were higher in 2015 than in 2004.

The study found that the overall prevalence of poor sleep was lower in 2015 than in 2004, and the 2015 prevalence was also lower in all subgroups than in 2004. The current study is consistent with previous studies in that females had poorer sleep quality than males [23,24,25]. It has been shown that, due to changes in estrogen and higher sensitivity to negative emotional information in females [26], psychological and physical pressures cause a higher probability of sleep and other related problems in females. Although the prevalence of poor sleep among older women in Shandong province has decreased over the past ten years due to the efforts of government departments and social factors, women still suffer from a poorer sleep quality than older men. Therefore, more social and policy support is required. In terms of regional differences, there was no statistically significant difference between urban and rural areas in 2004, while the prevalence of poor sleep was lower in urban areas than in rural areas in 2015. This may be due to the gaps between urban and rural areas in the economy, culture, education, medical security services, living environment, and convenience of living services [27]. With the advantages in the above aspects, older people living in an urban area may have a stronger sense of well-being and a higher quality of life [28], leading to fewer sleep problems compared to rural areas. In addition, due to the influx of young laborers from rural areas to urban areas during this decade, many elderly people were left behind in rural areas. Besides the above aspects, loneliness and a general lack of health awareness also contributed to poor sleep quality in the elderly living in rural areas [29]. The current study indicates that sleep quality and age had a negative relationship; the older the age, the higher the prevalence of poor sleep, which is consistent with previous findings [30]. One of the explanations for this is the reduction in melatonin release with age, which is produced by the pineal gland and contributes to sleep [31]. The reduction in deep sleep proportions in the elderly also causes a poorer sleep quality with age [32].

After more than 10 years of change, different characteristics contributed to poor sleep quality. In 2015, females, rural areas, living alone, higher age, lower level of education, and unemployment were the key factors associated with poor sleep quality, supporting the precise interventions for sleep problems in the elderly. Both regression models showed that the unemployed population is more likely to suffer from poor sleep, which may be due to a reduction in financial resources [33] and unaffordable living, inducing stress. In addition, as per capita GDP growth in China, the lower the income level, the worse the mental health [34], and the more likely to develop sleep problems. In 2015, the elderly living alone (including those who were divorced, unmarried, and widowed) became a new risk factor. Loneliness and a lack of social support could lead to mood disturbances, causing lower sleep efficiency and quality [35,36]. In addition, studies have shown that people who live with relatives are wealthier and live in better neighborhoods than those who live alone. The neighborhood environment significantly impacts physical health, including sleep [37]. Therefore, appropriate social support could alleviate sleep disturbances.

Through the results of our study, we found that the prevalence of poor sleep among the elderly in Shandong province is lower than the average in China [5]. Additionally, the prevalence of poor sleep is decreasing, indicating that economic and cultural development is conducive to helping older people improve their sleep. However, the prevalence of poor sleep is still higher among women, rural areas, and elderly groups with lower education levels. When formulating relevant public health policies, we can focus on the above and other vulnerable groups. Unfortunately, however, as the data were collected in 2015, some timeliness may be lost, which is a limitation we cannot avoid.

Since 1984, a province-wide epidemiological survey of mental disorders has been conducted in Shandong province every ten years [38]. Each epidemiological study of mental illness has provided a basis for decision making in the development of mental health work planning in Shandong province at that time, and the next survey is expected to be conducted in 2025. The sleep quality of the elderly is an essential component of the survey, and we believe our findings will contribute to the development public health and further research. Our first predictive model for poor sleep in the elderly provides an important reference for future related policy developments. In addition, the positive results of our study provide a reference for the next epidemiological survey on mental disorders.

## 5. Strengths and Limitations

Our study has the following strengths. First, the diagnostic criteria and study tools were consistent and comparable between the two surveys, and the results were reliable. Second, the use of the PSQI questionnaire allowed us to study not only the overall changes in sleep quality, but also each specific component. However, the study has several limitations. At first, we identified a large population sample of Shandong province, which is geographically and economically diverse and represents 9% of China’s elderly population, but our sample was not nationally representative. In addition, several variables are known to be associated with sleep quality in older adults, such as household income, mental health, and physical illness. Due to the limited data available, we were unable to account for these factors. Future research should address this issue and make the model more complete and more effective. In addition, the data were collected in 2015, which means it may have lost some timeliness. However, our results can provide a reference point for the next investigation. Furthermore, sleep quality was measured in this study using a retrospective self-report instrument (PSQI). Therefore, recall bias may have affected our findings. Future studies should use objective measures of sleep quality, such as polysomnography.

## 6. Conclusions

In summary, in theoretical terms, we confirmed that after more than a decade of effort, the overall sleep quality of elderly people in Shandong province has improved significantly. However, the incidence of sleep disorders in some groups, such as older women and rural areas, is still high. With the increasing aging trend in Shandong province, sleep problems in the elderly are still a problem that cannot be ignored. The risk factors that contributed to sleep quality were identified, which helps prevent and intervene in sleep disturbances, improving the sleep quality of the elderly population in Shandong province. It can also provide practical implications for the development of public health policy. Furthermore, the results of our study will provide an important reference for the next survey.

## Figures and Tables

**Figure 1 ijerph-19-14296-f001:**
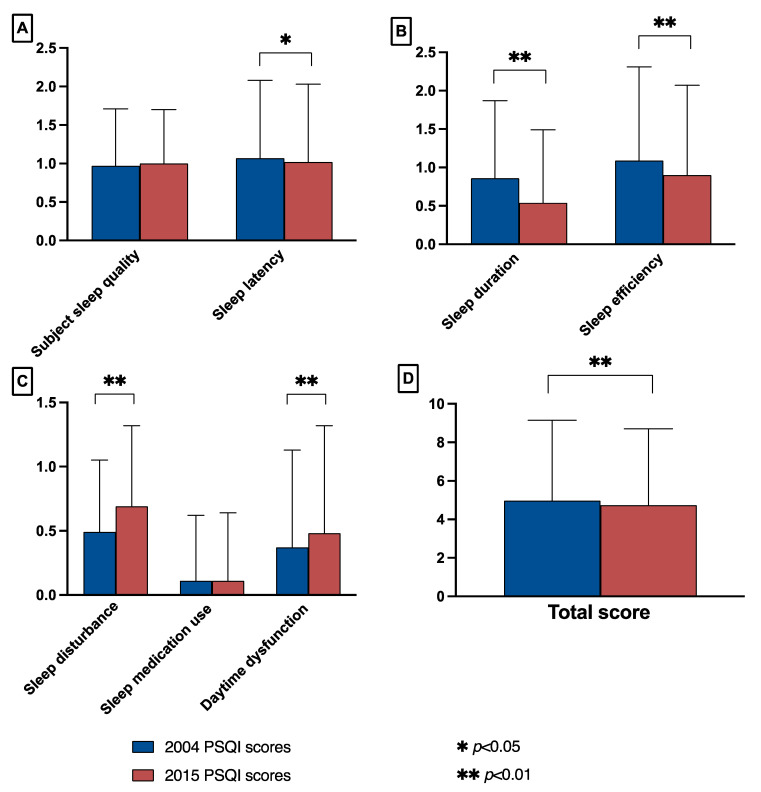
Comparison of PSQI scores for each factor between 2004 and 2015 for people aged ≥60 years in Shandong province. (**A**) The factors compared were: subjective sleep quality (effect sizes = −0.04), sleep latency (effect sizes = 0.05). (**B**) The factors compared were: sleep duration (effect sizes = 0.33), sleep efficiency (effect sizes = 0.16). (**C**) The factors compared were: sleep disturbance (effect sizes = −0.33), sleep medication use (effect sizes = 0.00), and daytime dysfunction (effect sizes = −0.13). (**D**) Comparison of total scores (effect sizes = −0.06).

**Figure 2 ijerph-19-14296-f002:**
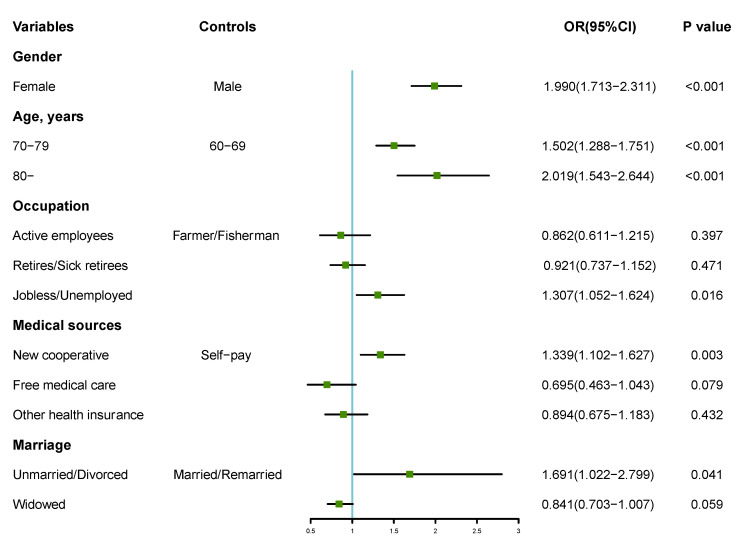
Forest plot of dichotomous multi-factor logistic regression analysis of factors influencing sleep quality among people aged 60 years and above in Shandong province in 2004. OR: odds ratio.

**Figure 3 ijerph-19-14296-f003:**
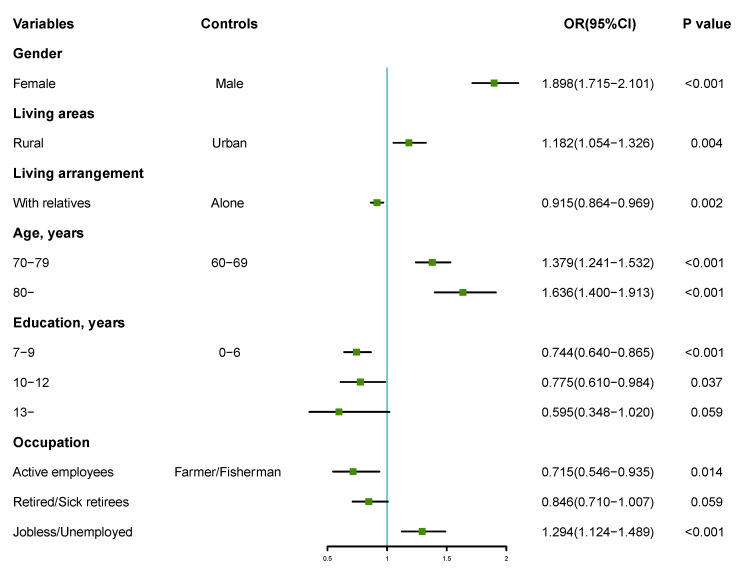
Forest plot of dichotomous multi-factor logistic regression analysis of factors influencing sleep quality among people aged 60 years and above in Shandong province in 2015. OR: odds ratio.

**Figure 4 ijerph-19-14296-f004:**
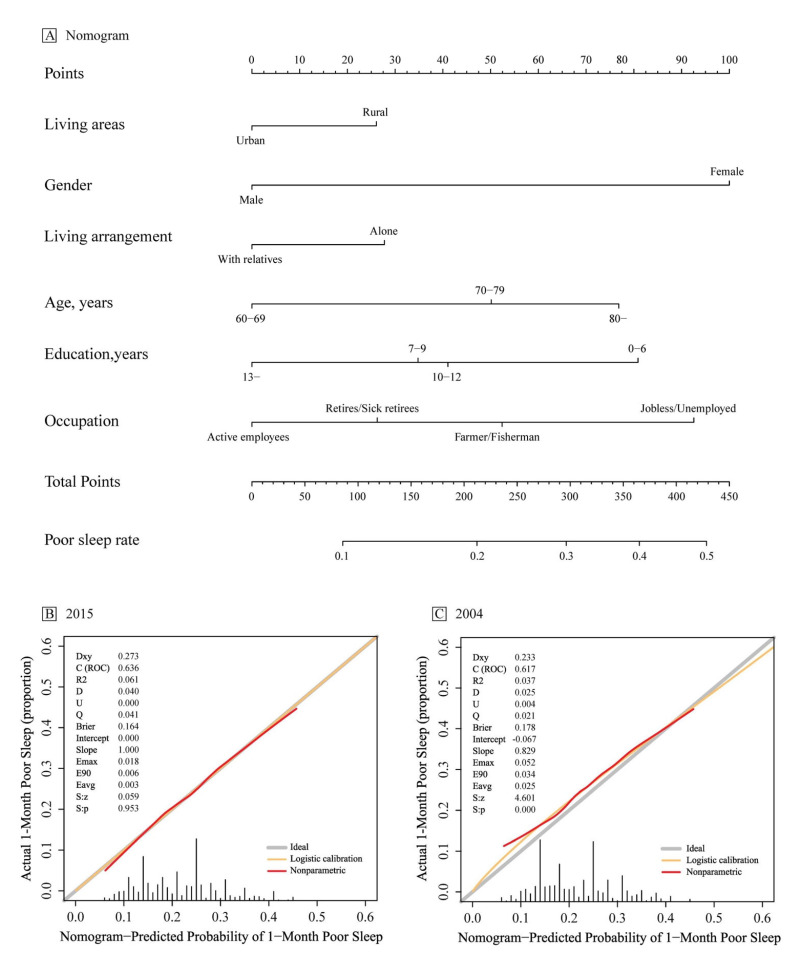
Nomograms for predicting the risk of poor sleep in an elderly population aged ≥ 60 years in Shandong province and its predictive performance. (**A**) Nomogram for estimating the risk of poor sleep in older adults in Shandong province. To use the nomogram, find the position of each variable on the corresponding axis, draw a line to the points’ axis for the number of points, add the points from all of the variables, and draw a line from the total points’ axis to determine the poor sleep probabilities at the lower line of the nomogram. (**B**) Validity of the predictive performance of the nomogram in estimating the risk of poor sleep presence in 2015 (*n* = 10,894). (**C**) Validity of the predictive performance of the nomogram in estimating the risk of poor sleep presence in 2004 (*n* = 4451). C-index, concordance index; U, calibration degree test.

**Table 1 ijerph-19-14296-t001:** Socio-demographic characteristics of the study population in 2004 and 2015, *n* (%).

Variables	2004	2015	*χ*^2^ Value	*p* Value
Gender			3.66	0.056
female	2324 (52.21)	5873 (53.91)		
male	2127 (47.79)	5021 (46.09)		
Living areas			134.87	<0.001
urban	1163 (26.13)	3905 (35.85)		
rural	3288 (73.87)	6989 (64.15)		
Age (years)			48.261	<0.001
60~69	2652 (59.58)	6508 (59.74)		
70~79	1500 (33.70)	3302 (30.31)		
≥80	299 (6.72)	1084 (9.95)		
Education (years)			71.50	<0.001
0~6	3530 (79.31)	8253 (75.76)		
7~9	574 (12.90)	1816 (16.67)		
10~12	223 (5.01)	667 (6.12)		
≥13	124 (2.79)	158 (1.45)		
Occupation			329.69	<0.001
farmer/fisherman	2446 (54.95)	7390 (67.84)		
active employees	231 (5.19)	496 (4.55)		
retired/sick retirees	1232 (27.68)	1691 (15.52)		
jobless/unemployed	542 (12.18)	1317 (12.09)		
Living arrangement			15.49	<0.001
alone	783 (17.59)	2219 (20.37)		
with relatives	3668 (82.41)	8675 (79.63)		
Medical sources			8371.98	<0.001
self-pay	3013 (67.69)	337 (3.09)		
new cooperative	636 (14.29)	8096 (74.32)		
free medical care	232 (5.21)	135 (1.24)		
other health insurance	570 (12.81)	2326 (21.35)		
Marriage			11.152	0.004
married/remarried	3315 (74.48)	8370 (76.83)		
unmarried/divorced	80 (1.80)	207 (1.90)		
widowed	1056 (23.73)	2317 (21.27)		
Total	4451 (100)	10,894 (100)		

*n*, total number of observations.

**Table 2 ijerph-19-14296-t002:** 1-month prevalence of poor sleep quality in a sample aged 60 years or older from Shandong in China between 2004 and 2015 (*n*, %).

Age (Years)	2004					2015					Standard Population	
	Total Number of Persons	Actual Number of Poor Sleepers	Before Adjustment Prevalence	Adjusted Number of Poor Sleepers	Adjusted Prevalence	Total Number of Persons	Actual Number of Poor Sleepers	Before Adjustment Prevalence	Adjusted Number of Poor Sleepers	Adjusted Prevalence	Number of People	Composition Ratio
60~64	1529	322	21.1	261	17.1	3616	630	17.4	527	14.6	73,382,938	27.8
65~69	1123	232	20.7	258	23.0	2892	594	20.5	625	21.6	74,005,560	28.0
70~74	965	268	27.8	233	24.1	2009	486	24.2	496	24.7	49,590,036	18.8
75~79	535	145	27.1	142	26.5	1293	343	26.5	341	26.3	31,238,849	11.8
≥80	299	103	34.5	209	69.8	1084	337	31.1	461	42.5	35,800,835	13.6
Total	4451	1070	24.0	1103	24.8	10,894	2390	21.9	2450	22.5	264,018,218	100

*n*, total number of observations.

**Table 3 ijerph-19-14296-t003:** Comparison of the 1-month prevalence of poor sleep between different populations aged ≥60 years in Shandong Province in 2004 and 2015 (*n*, %).

Variables	2004			2015			*χ*^2^ Value	Adjusted *p*-Value	Effect Size
	Total Number of Persons	Adjusted Number of Poor Sleepers	Adjusted Prevalence (95% CI)	Total Number of Persons	Adjusted Number of Poor Sleepers	Adjusted Prevalence (95% CI)			
Gender									
female	2324	718	30.9 (29.0~32.8) ^a^	5873	1671	28.5 (27.3~29.6) ^b^	4.81	0.028	0.07
male	2127	383	18.0 (16.4~19.6)	5021	788	15.7 (14.7~16.7)	5.83	0.016	0.10
Living areas									
urban	1163	263	22.6 (20.2~25.0)	3905	795	20.4 (19.1~21.6) ^b^	2.76	0.110	0.08
rural	3288	836	25.4 (23.9~26.9)	6989	1652	23.6 (22.6~24.6)	3.90	0.048	0.06
Age (years)									
60~69	2652	519	19.6 (18.1~21.1) ^a^	6508	1152	17.7 (16.8~18.6) ^b^	4.41	0.036	0.07
70~79	1500	375	25.0 (22.8~27.2)	3302	837	25.3 (23.9~26.8)	0.07	0.797	−0.01
≥80	299	209	69.9 (64.7~75.1)	1084	461	42.5 (39.6~45.5)	70.30	<0.001	0.69
Total	4451	1103	24.8 (23.5~26.0)	10,894	2450	22.5 (21.7~23.3)	9.33	0.002	0.23

*n*, total number of observations. ^a^, *p* < 0.001 is the chi-square test within the cohort in 2004. ^b^, *p* < 0.001 is the chi-square test within the cohort in 2015.

**Table 4 ijerph-19-14296-t004:** Comparison of actual mean sleep duration between different populations aged ≥60 years in Shandong province in 2004 and 2015 (h, mean ± *SD*).

Variables	2004	2015	*t* Value	Adjusted *p*-Value	Effect Size
Gender					
female	6.97 ± 1.85 ^a^	7.18 ± 1.94 ^b^	−4.87	<0.001	−0.11
male	7.50 ± 1.64	7.64 ± 1.76	−3.71	0.001	−0.08
Living areas					
urban	6.95 ± 1.52 ^a^	7.19 ± 1.65 ^b^	−4.58	<0.001	−0.15
rural	7.32 ± 1.85	7.50 ± 1.98	−5.25	<0.001	−0.09
Age (years)					
60~69	7.33 ± 1.69 ^a^	7.46 ± 1.78 ^b^	−3.78	0.001	−0.07
70~79	7.09 ± 1.87	7.31 ± 1.89	−3.91	<0.001	−0.12
≥80	6.97 ± 1.96	7.26 ± 2.31	−2.16	0.028	−0.13
Overall	7.22 ± 1.77	7.39 ± 1.88	−5.76	<0.001	−0.09

^a^, *p* < 0.001 for t-test or one-way ANOVA within the 2004 cohort. ^b^, *p* < 0.001 for *t*-test or one-way ANOVA within the 2015 cohort.

## Data Availability

The data presented in this study are available on request from the corresponding author. The data are not publicly available due to regulations of the ethics committees.

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
