# Peer review of "Sleep Quality among the Elderly in 21st Century Shandong Province, China: A Ten-Year Comparative Study"

_ijerph, 2022, doi:10.3390/ijerph192114296_

Round 1

Reviewer 1 Report

It's generally well described. However, there are some revisions and the contents are as follows.

<Materials and Methods>

1.     Describe securing representation of the study participants.

2.     Describe the reasons for using data from 2004 to 2015.

3.     Describe the evidence for selecting residents of China's Shandong region as study participants.

4.     Describe the rationale for the independent variables selected for application in this study.

< Discussion>

5.     Describe how the data applied to this study can be applied to today's environment.

Reviewer 2 Report

1- Introduction:

- The introduction needs more elaboration on the rationale for conducting this research.

- Another potential risk factors associated with sleep quality and well-being during the pandemic and here you may find some https://doi.org/10.1016/j.sleepe.2022.100030.

2- Method

- Please add the sub-section name study design and under this section, elaborate more on the study design implemented in this study with the rationale of selecting this design.

3- Discussion:

- In this section, you need to reflect on your work and evaluate the significance of the data obtained based on your experience, and it should include five main points: (1) a summary of the purpose/goals of the research, (2) theoretical implications of the findings, (3) practical implications of the findings, and (4) strengths, limitations, and recommendations for future studies.

4- Conclusions: Most needing reflection is the theoretical and practical implications.

Reviewer 3 Report

I like this paper a lot of for the quality of the content, quality of presentations and interest to the readers. For my side I do not have relevant comments in order to improve the paper. 

Author Response

Dear Reviewers:

Thank you for your great effort in reviewing our manuscript. Here, we have revised the manuscript based on your suggestions and provided a point-to-point response. We have highlighted changes to the manuscript in the document with tracked revisions and hope that our revisions and responses will address your questions and concerns.

Response to Reviewer 3 Comments

Point: I like this paper a lot of for the quality of the content, quality of presentations and interest to the readers. For my side I do not have relevant comments in order to improve the paper. 

Reply: We especially appreciate your appreciation of our manuscript and wish you all the best!

Reviewer 4 Report

- Introduction: Good synthesis and explanation of the study problem.

- Methods: correctly performed. 

- Results and discussions are okey. 

From my point of view, all the relevant fields of an investigation of this type are well developed.

Author Response

Dear Reviewers:

Thank you for your great effort in reviewing our manuscript. Here, we have revised the manuscript based on your suggestions and provided a point-to-point response. We have highlighted changes to the manuscript in the document with tracked revisions and hope that our revisions and responses will address your questions and concerns.

Response to Reviewer 4 Comments

Point: - Introduction: Good synthesis and explanation of the study problem.

- Methods: correctly performed. 

- Results and discussions are okey. 

From my point of view, all the relevant fields of an investigation of this type are well developed.

Reply: We thank you for your kind words about our manuscript and we wish you good health and a happy life!

Reviewer 5 Report

The authors presented the results of ten-year comparative study of sleep quality among the elderly in 21st century Shandong Province, China. This is an interesting topic, but I have some comments:

1) Change the format of Figure 1 into a boxplot. Description of data format must be added to legend of this Figure.

2) Quality of Figure 4 is low. The resolution must be increased.

Otherwise, it is a quality study with interesting results.

Reviewer 6 Report

After reading the manuscript entitled “Sleep quality among the elderly in 21 st century Shandong province, China: A ten-year comparative study” I made some remarks, which I hope will help the authors to improve their manuscript.

Overall, I think that methodologically the study is quite well done. However, the authors need to review mainly their introduction, their discussion but especially the quality of the presentation of their figure.

- The discussion session is missing in the abstract

- The introduction is very focused in China. It is essential that the authors start from the situation of global aging in the world before addressing the situation in China. In other words, the authors lack contextualization of aging in the world. This work is essential to give more consistency to the introduction.

- Line 76 : Authors must specify the agreement number of the Institutional review board approved the study.

- The first line of tables 2 and 3 must be revised as they are unreadable.

- Throughout the document the p of significance must be in lower case and italics. Also, the authors should specify the effect size of significant results.

- Line 91 : replace "Liu et al" by "Liu and collaborators". Search the entire document for similar errors and correct

- Figure 4 is unreadable

- I suggest that the authors write a paragraph in the discussion about the research perspectives that arise from the results of their studies.

- Also in the discussion, I would like to know if the authors controlled for melatonin levels in the participants. Indeed, the authors claim that this hormone plays a role in the negative relationship that would exist between sleep quality and age.

- Line 279 : remove “2.Materials and Methods

- The authors' contributions are not shown at the end of the document.

In short, authors still need to improve the quality of their manuscript. Finally, it is a bit unfortunate that the authors do not have an intermediate measure between 2004 and 2015

Reviewer 7 Report

In general, the article deals with an interesting problem, is well structured and reaches conclusions that, in principle, can be described as useful. The statistical techniques used are correct and have been put to good use.

However, the most recent data with which we work is from 2015 and the topic addressed is likely to depend on and vary over time, and up to the present 2022, a sufficient number of years have elapsed to cast doubt on whether the conclusions obtained are true today. This is a situation that underlies the thinking of the authors, given that they state that from 2004 to 2015 (period studied) the characteristics of the population studied have changed significantly. Is it possible that they have changed from 2015 to 2022?

MINOR CONSIDERATIONS

Some minor considerations that it would be opportune for the authors to attend to, even though they are formal, would be the following:

- The two samples used are of considerably different sizes: 4,451 in 2004 and 10,894 in 2015. However, the reason why this difference appears is not justified in any case. I consider it opportune to dedicate a brief comment that justifies it.

- Among the tools used for statistical analysis, two versions of SPSS are mentioned: SPSS 15.0 and SPSS 26.0. Why are two versions of the same software used? What does the old version (SPSS 15.0) provide over the newer version (APSS 26.0)?

- The graphics included in the article (figures 2, 3 and 4) and the tables have been little taken care of, so it would be convenient to redo the graphics looking for a better definition and take care of the legends of the tables.

Round 2

Reviewer 6 Report

Seeing the corrections made by the authors, I find the article worth accepting. Good job.